# Electrochemical Synthesis of Unique Nanomaterials in Ionic Liquids

**DOI:** 10.3390/nano11123270

**Published:** 2021-12-01

**Authors:** Olga Lebedeva, Dmitry Kultin, Leonid Kustov

**Affiliations:** 1Department of Chemistry, Lomonosov Moscow State University, 119991 Moscow, Russia; lebedeva@general.chem.msu.ru (O.L.); dkultin@general.chem.msu.ru (D.K.); 2N.D. Zelinsky Institute of Organic Chemistry, Russian Academy of Sciences, Leninsky Prospect 47, 119991 Moscow, Russia; 3Institute of Ecology and Engineering, National Science and Technology University “MISiS”, Leninsky Prospect 4, 119049 Moscow, Russia

**Keywords:** ionic liquids, nanomaterials, nanostructure, electrochemistry, electrosynthesis

## Abstract

The review considers the features of the processes of the electrochemical synthesis of nanostructures in ionic liquids (ILs), including the production of carbon nanomaterials, silicon and germanium nanoparticles, metallic nanoparticles, nanomaterials and surface nanostructures based on oxides. In addition, the analysis of works on the synthesis of nanoscale polymer films of conductive polymers prepared using ionic liquids by electrochemical methods is given. The purpose of the review is to dwell upon an aspect of the applicability of ILs that is usually not fully reflected in modern literature, the synthesis of nanostructures (including unique ones that cannot be obtained in other electrolytes). The current underestimation of ILs as an electrochemical medium for the synthesis of nanomaterials may limit our understanding and the scope of their potential application. Another purpose of our review is to expand their possible application and to show the relative simplicity of the experimental part of the work.

## 1. Introduction

“Nanotechnology and nano-engineering are two opposite sides of a visionary coin. On the one hand, scientific revelation, deep scientific provenance and scientific ingenuity are the pivots of scientific research pursuit in nanotechnology and nanomaterials today. Frequent environmental disasters, global climate change, and stringent environmental regulations are challenges for efficient and scientifically sound solutions. Nanotechnology is deeply integrated with diverse areas of science and engineering, due, in part, to ever-growing concerns about global warming and climate change” [1]. Nanomaterials can potentially find applications in various fields of science and technology. Nanotechnology and nanomaterials have already found their way into numerous consumer products we use in everyday life, from clothing to skin lotion [2]. Nanomaterials are widely used in medicine [3], the textile and food industries [4], the production of cosmetics and environmental protection [5]; however, their potential has not yet been explored in full.

The properties of nanoparticles and the possibility of their use for solving practical problems depend on their size and the state of their surface; therefore, these are to a significant extent determined by the methods of their production [6].

Among the chemical, physicochemical, physical and mechanical methods for producing nanoparticles [7], electrochemical methods based on the use of ionic liquids deserve special attention. The synthesis of self-organized nanostructures in ionic liquids is facilitated by the fact that ionic liquids themselves, as it will be shown below, are characterized by a certain degree of structural organization.

The electrosynthesis of any material depends on the electrochemical window of the electrolyte, the electrode material, the temperature, the nature of the precursor, additives and basic electrochemical parameters [8]. Electrochemical methods make it possible to carry out synthesis in one stage and, by varying the experimental conditions, to obtain nanostructures of various sizes, shapes and compositions.

Ionic liquids (ILs), i.e., low-temperature melts of salts with, as a rule, bulk organic cations and inorganic/organic anions, have already proven themselves as systems that are promising in various fields of applications. Since ionic liquids consist almost entirely of “free” charge carriers—cations and anions—their application in electrochemistry is especially interesting. ILs have a sufficiently high ionic conductivity [9,10], and therefore their use in electrochemistry, electrocatalysis and electroplating is of undoubted industrial interest. The uniqueness of ionic liquids is associated with their high electrochemical stability, relatively high electrical conductivity and the absence of a measurable saturated vapor pressure. Of course, a significant amount of data should be accumulated to claim that electrochemical processes in ionic liquids unequivocally fit into the sphere of “green” chemistry. It is known, for example, that ionic liquids with the PF_6_^−^ anion are not completely inert, including in their application in electrochemical processes, and the use of such ionic liquids is now limited. Nevertheless, in some cases, ionic liquids demonstrate significant advantages over traditional electrolytes. Currently, ionic liquids with an exceptionally large electrochemical window (5–9 V) have appeared, and they can be used repeatedly without any loss of properties or destruction of their structure. The prospects for using ionic liquids in the processes of electrodeposition, electroplating and electropolishing of metals and alloys and in the processes of preparation of metal nanoparticles and their alloys, as well as oxide nanostructures, by electrochemical methods are attractive.

The purpose of this review is to demonstrate the possibilities of electrochemical methods for the synthesis of diverse nanostructures using ionic liquids as electrolytes and stabilizers of nanoparticles. Another goal of the review is to consider an aspect of the applicability of ILs that is usually not fully reflected in modern literature, the synthesis of nanostructures (including unique ones that cannot be obtained in other electrolytes). The current underestimation of ILs as electrochemical media for synthesis may limit our understanding and their potential application. To expand the possible application and to show the relative simplicity of the experimental part of the work is another purpose of our review.

The chemical composition of the obtained nanomaterials can be chosen as the basis for the structure of the review: nanoparticles based on carbon (fullerenes, nanotubes, graphene) and silicon; nanoparticles of metals, oxides, etc.; and nanofilms of organic polymers. Further classification can be pursued on the basis of the dimension [11,12,13]:zero-dimensional (0D) particles (e.g., quantum dots or core-shell particles);one-dimensional (1D) moieties (nanowires; nanorods; nanotubes, including carbon and oxide materials; nanoribbons, etc.);two-dimensional (2D) materials (flat structures, self-organized monolayers, nanodiscs, graphene);three-dimensional (3D) structures (dendritic structures, etc.).

The tables of designations of cations (Table 1) and anions (Table 2) of ILs mentioned in the review is presented below.

## 2. Peculiarities of the Processes of Electrochemical Synthesis of Nanostructures in Ionic Liquids and the Methods Used

The available reviews [12,14] provide general characteristics of dimensional effects, the systematization of dimensional effects in electrochemical systems, the prospects of electrode processes and nanoelectrochemistry. Nanoelectrochemistry solves three interrelated tasks, namely:preparation of nanomaterials with a given structure and properties;the use of nanomaterials for a specific purpose, taking into account their structure and properties;control of the structure and properties of nanomaterials both during their production and in the course of their application.

Electrodes with characteristic dimensions on the order of about 10 nm are sometimes called nanoelectrodes or nanodes [12,15,16]. Nanodes can be prepared from nanoscale materials with a known structure, for example, nanotubes. The development of various research methods, including scanning electrochemical microscopy (SECM) and scanning tunneling microscopy (STM), have served as an additional impetus to the development of nanoelectrochemistry [12].

The nanoscale effects in electrochemistry can occur not only due to the electrode, but also at the electrode–electrolyte interface, namely due to the existence of a double electric layer. The spatial interphase boundary arises as a result of electrostatic and chemisorption interactions. The zero charge potential of the metal plays an important role in the formation of a double electric layer [17], with the value of this potential being determined not only by the chemical nature of the metal, but also by its crystallographic orientation. At a certain thickness of the electrode films, it is possible to achieve a dimension-controlled dependence of the electrode charge on its potential. In general, the composition of the double electric layer may differ greatly from the bulk composition, while the state of the particles forming the double electric layer may also change. A well-known example of such a change is the specific adsorption of ions or the formation of submonolayers of adatoms stabilized by interaction with the substrate. The latter phenomenon is called underpotential deposition (UPD), since precipitation occurs before reaching the equilibrium potential for this redox system [18]. The properties of the layer become similar to the properties of the bulk phase only when the fourth layer is deposited.

Size-dependent electrochemical phenomena are observed in chemisorption and electrocatalytic processes occurring on highly dispersed electrodes with a nonhomogeneous surface. The sizes of inhomogeneities in such processes are commensurable with the sizes of the adsorption layers—namely, they correspond to the nanoscale and therefore they become essential. Many electrodes exhibiting catalytic properties are so-called nanostructured materials characterized by periodic heterogeneity not only at the surface, but also in the bulk. Nanostructures are most often nonequilibrium but relatively long-lived formations [12,14].

There are two main directions in the electrochemistry of nanoparticles: the production of electrodes with deposited nanoparticles and electrochemical methods for the production of nanoparticles [19].

Methods for the preparation of nanoparticles on conductive substrates can be divided into three groups:simultaneous formation and immobilization of nanoparticles;immobilization of metal ions with subsequent reduction to metal nanoparticles;preparation of metal nanoparticles with their subsequent immobilization on a substrate.

Electrochemical methods are effective for conductive and semiconductive substrates. In general, electrochemical methods are considered as a case of liquid-phase chemical methods with the participation of electrons [12]. supplied from an external electrical circuit. Electrochemical methods can be implemented in the “bottom-up” version, with atoms or molecules acting as precursors, and “top-down” version, when nanostructures are obtained from a compact material [20].

Electrodeposition is one of the most advanced areas of electrochemistry, which is successfully used for the synthesis of nanomaterials based on metals [21], alloys [22], oxides [23], conductive polymers [24] and other nano-sized materials [25,26,27,28].

Galvanostatic, potentiostatic and potentiodynamic [29] pulse alternating-current [12,30] modes are used for the electrodeposition process. The use of these methods makes it possible to govern the synthesis of nanoparticles by changing the composition and viscosity of the electrolyte, by varying the process time, as well as by changing the potential or the current density. The significant advantages include the possibility of conducting the synthesis in one stage and without using templates. Metal or alloy nanoparticles can be deposited on the surface of a conductive or semiconductive substrate serving as a cathode, which allows the obtaining of electrodes for further use. Most often, polydispersed samples are obtained in this way. To obtain electrodes with a narrow size distribution of nanoparticles, it is important to know the mechanism of formation of nuclei on the surface, for which the Sharifker–Hills model can be used [31].

Nanodes are used for a more detailed study of the nucleation and growth of nanoparticles. Electrodeposition can be attributed to the “bottom-up” technology.

Template electrodeposition is a version of conventional electrodeposition. The advantage of this method is that the process is carried out under milder conditions compared to conventional electrodeposition. The morphology of the nanomaterial is uniquely determined by the shape and size of the template pores and the amount of electricity passed through. In addition, two or more components can be deposited on the membrane to form a hybrid material [12]. The scheme of template electrosynthesis of nanowires includes the deposition of the template on the working electrode, the electrodeposition of metal in the pores of the template and the subsequent dissolution of the template [32].

The methods of open-circuit autocatalytic deposition and galvanostatic (spontaneous) substitution related to electrodeposition are used in the synthesis of core–shell nanostructures [12,14].

Nanostructures on the surface can be obtained by “top-down” technology, the anodic processes occurring at the same time are attributed to the processes of etching, polishing and anodizing [33]. The following factors influence the result of the anode effect: the current density, voltage, composition, concentration and viscosity of the electrolyte, the ratio of the anode–cathode area, the temperature and the state of the anode surface [34]. Despite the widespread use of these methods for the synthesis of various materials, there is still no complete understanding of the electropolishing process. In general, it is believed that two separate reactions occur on the surface of the anode, called smoothing and gloss formation. The rate of dissolution in the recesses and at the tops depends on the current distribution or mass transfer conditions. This process (smoothing) usually leads to a decrease in roughness at the micron level and can be controlled by: (a) ohmic drop, (b) activation and (c) mass transfer in the metal dissolution reaction. Anodic gloss formation can be achieved only if the dissolution of the metal is controlled by mass transfer, when it is possible to form a salt deposit on the surface (i.e., the formation of a viscous layer). The mechanism of self-organization during the electropolishing of metals in aqueous solutions, which leads to the formation of nanostructures on the metal surface, is considered in detail in the works by Yuzhakov et al. [35,36]. This model is based on the Debye–Hückel approximation used for diluted electrolytes. By sequentially solving the equation for a flat surface and for the electrode configuration deviating from a flat one, the first-approximation correction to the dissolution rate is obtained. If the first-approximation correction is greater than zero, the solubility increases on the protrusion of the surface, which corresponds to the negative feedback characteristic of standard electropolishing. In the case of the positive feedback, the protrusions will dissolve more slowly and structures will appear, for example, in the form of hexagonal cells. The authors suggest that the mechanism of instability is based on the adsorption of molecules causing a “shielding” effect, preferably on the protrusions of the surface, thereby preventing dissolution [37].

The influence of the state of the metal surface on the formation of nanostructures in the form of cells on the electrode surface has been demonstrated [35,36].

When using suitable ionic liquids (ILs), reagents and products that are unstable in other solvents can become stable and redox processes that do not occur in a molecular solvents are possible. The range of reaction conditions becomes noticeably wider in ILs compared to traditional solvents. Some fundamental electrochemical concepts commonly used for standard solvents are not yet always suitable for ILs [38,39].

It is usually stated that ILs consist of simple ionic particles, are non-flammable, non-volatile, highly thermally and electrochemically stable, and are sometimes referred to as part of “green chemistry.” However, over the past few years, both as a result of the increasing knowledge in the field of physical and chemical properties and also as a result of the rapidly expanding range of available ILs, it has become clear that, in fact, none of these general properties are fully applicable to ILs. An urgent question arises: what is the universal property of ILs? MacFarlane believes [40]: “If we accept the definition of an ionic liquid as being an ionic compound (a salt) which is liquid below 100 °C, then in fact the only defining properties that one can expect to observe, a priori, are that, at some temperature below 100 °C: (a) the substance is liquid (its glass transition temperature and/or melting point are below 100 °C), and (b) it contains ions and therefore exhibits ionic conductivity.”

Therefore, a certain level of ionic conductivity is the only universal feature that can always be expected in an ionic liquid.

Ionic liquids are often called “designer solvents” due to the possibility of rational selection and design of the chemical structure of the cation and anion, which provides an optimal structure–property relationship [41,42]. The main mechanisms of electrochemical behavior of cations in ILs are considered [43,44,45]. According to the characteristic of cathodic stability, ILs cations can be arranged in the following order: pyridinium < sulfonium < 1,3-dialkylimidazolium, <1,2,3-trialkylimidazolium < morpholinium < phosphonium < quaternary ammonium. For phosphonium and quaternary ammonium ions, the cathode limit is −2.7 V. The oxidation stability of anions can be represented as: [TFSA]^−^ > [FAP]^−^ > [TfO]^−^ > [DCA]^−^ > [TFA]^−^. These data were obtained for a glassy carbon working electrode [44]. The properties of ionic liquids as electrolytes, a dispersion medium, as well as the use of ILs in the synthesis of nanoparticles are discussed in a number of reviews [40,41,42,43,46,47,48,49,50,51]. The reactivity and reaction rate for ILs can be properly explained on the basis of the theory of linear solvation energy relations using a combination of Camlet–Taft polarity parameters. In this case, the reactivity can be predicted equally for both ILs and conventional solvents [52]. Another point of view takes into account the supramolecular nanostructures of ILs, paying special attention to ILs consisting of aromatic ions. From this point of view, the degree of structural organization of ionic liquids plays a significant role, along with specific weak interactions between IL ions and reagents and transition states.

It should be noted that deep eutectic solvents (DES) are now widely recognized as a new class of analogs of ionic liquids (ILs), since they have many common characteristics and properties. In the literature, the terms DESs and ILs are often used interchangeably [53], although it should be noted that according to some authors [54], these are two different types of solvents. A somewhat more detailed discussion of DESs is given at the end of this section.

The processes of synthesis of nanostructures (particles, films, tubes, etc.) are determined by the self-organization of the system, including the ILs itself, the electrode, and the precursor of the nanomaterial (if the nanomaterial is not obtained from the electrode material). Various self-organized structures based on nanoparticles in ionic liquids are formed as a result of a balance of various intermolecular interactions (Van der Waals, electrostatic, hydrogen bonds, π–π interactions, etc.). The structure of the ILs was initially established based on the structure of crystal analogs. It was assumed that the arrangement of the ions is isotropic, with some similarity to the solid or liquid crystal state [50]. Thus, the melting process for ionic liquids seemed similar to molecular liquids, where long alkyl chains in the substituents in the ILs cation can take different conformations, which leads to the formation of crystalline polymorphic forms. For example, the X-ray diffraction method shows the existence of diverse conformations of butylmethylimidazolium [BMIm]^+^ cations with Cl^−^, Br^−^, I^−^, [BF_4_]^−^ and [PF_6_]^−^ anions. Depending on the conformation of the butyl group in the [C_4_MIm]^+^ cation, ILs can be crystallized either in a monoclinic or orthorhombic forms. Both forms are in equilibrium in the liquid state. Anions can also have different conformations. In particular, the widely used [Tf_2_N] anion in the crystalline phase is represented by a cis-conformer, and in the liquid phase it mainly exists in the trans-form [50].

Figure 1 shows the scheme of domain formation in the ionic liquid 1-hexyl-3-methylimidazolium hexafluorophosphate [C_6_MIm][PF_6_]. The red color indicates the imidazolium cation and the hexafluorophosphate anion, the green color indicates nonpolar alkyl substituents in the imidazolium ring. For ionic liquids based on the imidazolium cation with long alkyl substituents, the chains of alkyl groups can be distinguished in the form of nonpolar domains, while other parts of the ionic liquid can be distinguished in the form of polar domains, as shown in Figure 1. Increasing the length of alkyl substituents increases the nonpolar domains, which increase in size and become more connected, which leads to microphase stratification, similar to the formation of a liquid crystal.

The distribution of a dissolved substance in ILs is very different from the distribution in homogeneous solvents due to the existence of structural inhomogeneities in ILs. Solutes tend to be localized in those nanoscale domains to which they have a high affinity [48].

Due to the variety of ions that form ILs and have different abilities for self-assembly, self-organization phenomena are observed both in the volume of the ionic liquid itself and at the phase interface [51].

In the case of the use of ionic liquids in the synthesis of nanoparticles and other nanostructures, the number of stages and reagents used is significantly reduced, since ILs are simultaneously stabilizers of nanoparticles [55,56]. The role of electrostatic interactions in the stabilization of nanoparticles is currently recognized [48,56,57,58,59]. Cations of ILs are attracted to the surface of a negatively charged nanoparticle to form a positively charged ion layer, and then the counterions form a second layer on the surface of the nanoparticle. It was shown that anionic supramolecular aggregates of the composition [(BMIm)_−n_(PF_6_)_x_]^n−^ form the first layer on gold nanoparticles with a diameter of 2.6 ± 0.3 nm. Supramolecular cationic aggregates of the composition [(BMIm)_x_ (PF_6_)_x−n_]^n+^ provide the charge balance (Figure 2) [48].

It was possible to establish the presence of near-surface layers of ILs using atomic force microscopy [60]. A larger number and more strongly bound layers are formed on the charged surface. Since the boundary structures are formed in different ways depending on the surface charge, the boundary order of ions changes the volume flow on an uncharged surface (mica) and highly oriented pyrolytic graphite (Figure 3).

It was shown [61] that near the neutral surface of graphite, the structure of the ionic liquid differs from the structure in the bulk, and is a quasi-crystalline phase with a length of ~1.5 nm, characterized by low ion mobility due to their spatial and orientation ordering. Parallel layers of increased density are observed, consisting of an accumulation of electrostatically bound anions and positively charged imidazole rings. The ions adsorbed on the surface form two-dimensional molecular clusters. Two types of clusters are observed. In the first type, anions are self-organized in the form of fragments of a triangular lattice containing about 5–10 ions, while the subsystem of cations is disordered. In the second type of clusters, both types of ions are present, which together form a fragment of the hexagonal lattice.

Silicon (Si) is currently considered as a promising material for the negative electrode in Li batteries, due to its much higher specific energy compared to graphite. However, the use of silicon is limited by the unstable interphase boundary of the electrode/electrolyte (SEI solid/electrolyte) interface. Silicon particles undergo volumetric changes during the battery operation, as a result of which the particles crack and new areas of the surface are exposed, where electrochemical decomposition of electrolyte components occurs. Using experimental and computational methods, the influence of the nature of the ionic liquid cation and the concentration of a lithium salt on the interfacial structure of the Si electrode and SEI formed during polarization was revealed [62]. The study shows a clear relationship between the nano-structure of super-concentrated IL based on P1222FSI near the electrode surface, the electrode/electrolyte (SEI solid/electrolyte) interface and the cycling characteristics of the negative electrode Si.

The size, size distribution and morphology of the nanoparticles synthesized in IL depend on the physicochemical properties of the ionic liquid, which affect the stabilization of the nanoparticles. A smaller diameter and a narrower size distribution of nickel and zinc oxide nanoparticles are obtained in ionic liquids with a longer substituent at the cation [48].

As already shown above, the peculiarity of nanoscale effects in electrochemistry is that these effects can occur not only at the expense of the electrode, but also at the electrode–electrolyte interface, namely due to the existence of a double electric layer. Traditionally, the double electric layer (DEL) is described using the classical Gui–Chapman model developed for dilute electrolytes. The expansion of the use of IL served as an impulse to create a new picture of DEL in relation to very concentrated ionic systems, since the classical model is not able to describe them. Theories and models describing the structure of DEL in an ionic liquid have been proposed; however, a complete description of the double layer at the molecular level is still required. Ionic liquids form alternating layers of anions and cations with a length of several nanometers, similar to the phase of a liquid crystal.

ILs are liquid salts with a traditional Debye length of less than atomic dimensions. This means that they behave like molecular liquids and are not expected to show a continuous ionic response to a change in the electric field. The key feature of this behavior is the formation of a purely molecular layer on the surface of a solid substrate. This can occur through a maximum increase in the density of ion packaging when the surface electrode potential is applied. It has been shown that the formation of a densely packed monolayer for [DCA]^−^ and [BF_4_]^−^ anions is possible only at surface charges exceeding the limit of the electrochemical stability of the corresponding ionic liquids. For the [TBA]^+^ cation, a monolayer structure can be observed at a charge of about 30 µQ/cm^2^ achieved in an electrochemical experiment [63].

To date, we can unequivocally state [48]:The dependence of the DEL capacity on the electrode potential is bell-shaped, and there are no signs of compliance with the Gui–Chapman theory [64];Significant overscreening effects (excessive shielding) are observed at small electrode polarizations;At large polarizations, the effect of “lattice saturation” (maximum increase in ion packing) is observed;The thickness of the DEL in IL is larger than one layer of particles.

An attempt has been made [65] to model the distribution of IL particles not only along the normal to the surface of the graphite electrode, but also along the plane of the electrode, which makes it possible to create a 3D model of a double electric layer.

According to Gomez et al., the molecular structure of the ionic liquid and the influence of the surface-induced orientation should include complex Coulomb, Van der Waals, dipole interactions, hydrogen bonds and steric interactions [66].

Imidazolium-based ILs can interact with many substances, since they have both hydrophobic and hydrophilic fragments in their composition and are highly polarizable. This specific property of imidazolium ILs distinguishes them from classical ionic aggregates, in which the formation of ion pairs and even ion triplets is widely recognized. Such an organization of the ILs structure can be used as an “entropy initiator” in the synthesis of spontaneous, well-organized and ordered nanostructures. Indeed, the unique adaptability to other molecules and phases, combined with the possibility of forming strong hydrogen bonds, makes ILs a tool in the preparation of a wide range of new generations of nanostructures [67].

Ionic Liquids and Deep Eutectic Solvents

The term ionic liquid was initially rigorously applied to systems that only contained ionic species which melted below 100 °C [40], but this definition has slowly been relaxed to include a wider range of fluids where ionic character tends to dominate solvent–solute interactions. “The field of ionic liquids can become a much broader field of low melting salts–we leave the word low intentionally undefined and purposely avoid labelling these as organic vs. inorganic salts. This opens up perspective on this area of chemistry to an even more massive family of potentially interesting compounds” [68].

Deep eutectic solvents are just one of these ion-dominated systems which have been found to be useful for a variety of applications. While the first definition of DESs involved quaternary ammonium salt and hydrogen bond donor mixtures, they can be more generally described as mixtures of Brønsted and Lewis acids and bases [69].

Although ILs and DESs have a lot in common, especially when it comes to physical properties as well as applications, from the chemical point of view, these are two separate groups of substances [70]. ILs are an ***organic molecule*** possessing an ionic part while DESs are ***salt mixtures*** (Figure 4).

DESs can be made from components that have renewable sources, and DESs composed entirely of plant metabolites (such as ammonium salts, sugars and organic acids) are labeled natural deep eutectic solvents (NADESs). These solvents have low vapor pressures, good thermal stabilities and broad applications, which make them extremely attractive as replacements for traditional organic solvents. The cost of production is low due to the low prices of synthetic raw materials.

However, concerns arise particularly for soil and water contamination at the point of disposal. DESs should not automatically be assumed to be nontoxic if they are prepared from naturally occurring compounds. The use of DESs is not without challenges. For example, oil extractions cause small amounts of oil to be retained in the DESs [42]. In some cases, high viscosity limits the use of DESs [69]. Unfortunately, their high viscosity and solid state at room temperature could be detrimental. However, the physicochemical properties of DESs can be tailored by the selection of proper HBA (hydrogen-bond acceptor) and HBD (hydrogen-bond donor) and their molar ratio or the addition of water.

Very few studies have considered DES–electrode interfaces and it remains unclear whether classical theories of electrical double layers capture the behavior of DESs, or if they are better described by the developing theories for concentrated electrolytes such as ILs [72]. Authors have represented the general schematic of critical ion interactions in choline chloride:ethylene glycol (ChCl:EG) systems for a metal and non-metal electrode surface. The obtained results indicate DESs and H-bonded electrolytes with high salt concentrations are distinct from ionic liquids and present no pronounced overscreening or crowding with applied potential.

## 3. Production of Carbon Nanomaterials

There are few well-described examples in the literature of the application of electrochemical methods for the production of carbon nanomaterials using ionic liquids as media and electrolytes. Nevertheless, electrodeposition methods have been used, including electrochemical reduction of chlorine-containing compounds, including carbon tetrachloride CCl_4_, which lead to the formation of sufficiently thick graphite films on the surface of a nickel electrode, while the formation of polyacetylenes was noted in the presence of small amounts of water [73].

These studies are generally consistent with the data obtained in studying the processes of synthesis of carbon nanomaterials in salt melts [74]. Ferromagnetic carbon nanotubes were obtained by the electroreduction of CO_2_ in molten salts with anodes made of steel or nickel alloys [75].

In these works, the possibility of the one-stage production of graphite, graphene, ultrafine amorphous carbon, doped and undoped carbon nanotubes and fibers both in the form of films and in the form of fine powders has been demonstrated. It should be noted that carbon dioxide dissolved in salt melts can act as a precursor of carbon nanomaterials. This process was investigated by the method of cyclic voltammetry, and the mechanisms of processes occurring in the cathode and anode regions have been proposed. The influence of the process conditions (electrolyte composition, temperature, cathodic current density, potential) on the synthesis efficiency has been studied. The structure of the resulting carbon nanomaterials has been established by X-ray diffraction, electron microscopy and Raman spectroscopy. Carbon nanomaterials modified by IL have the advantages of both components of the functionalized material [76].

## 4. Silicon and Germanium Nanoparticles

Nanostructured semiconductors attract attention due to their unique and tunable electronic and optical properties and potential use in nanosensors, catalysis, solar cells, batteries and biomedicine. The synthesis, application and properties of nanosilicon are described in detail in the recent monograph [77]. The production of one-dimensional (1D) semiconductor nanostructures grown on two-dimensional (2D) nanomaterials for use in flexible electronic and optoelectronic devices [78] and the use of silicon nanostructures as electrodes for microcondensors are promising [79]. Ionic liquids have found application in the electrochemical synthesis of various semiconductor nanostructures.

Silicon and germanium nanowires were obtained for the first time electrochemically in the ionic liquids [BMPy][NTf_2_] or 1-hexyl-3-methylimidazolium bis(trifluoromethanesulfonyl)imide containing SiCl_4_ and GeCl_4_ as precursors using the template deposition technique [80,81,82]. Commercially available polycarbonate membranes with an average pore diameter of 90 nm or polystyrene substrates were used as templates. One side of such a membrane was covered with a 120 nm thick gold film and acted as a working electrode during the electrochemical experiment. Electrodeposition was performed at room temperature inside the pores. The obtained silicon and germanium nanowires were amorphous, cylindrical, smooth and uniform in diameter, as determined by the diameter of the membrane pore. Annealing makes it possible to crystallize silicon nanowires without changing their shape and composition [81]. It was shown that nanowires of the composition Si_x_Ge_1−x_ can be obtained in ionic liquids [BMPy][NTf_2_] and [EMIm][NTf_2_] containing both SiCl_4_ and GeCl_4_ by electrochemical template deposition [83]. Amorphous silicon nanocolumns were deposited in a template of trimethylhexylammonium bis(trifluoromethanesulfonyl)imide([N_1116_][NTf_2_]) containing SiCl_4_ [84]. Template-free electrosynthesis of Si–Sn nanowires in ionic liquids is described in the recent review [8].

In one of the recent reviews [85], analysis of all works on the synthesis of silicon nanostructures by electrochemical methods using various salt melts, including ionic liquids, has been carried out. It is noted that the use of electrochemical methods makes it possible to abandon the use of harmful reducing agents, since electrons act as a reducing agent. Thus, a one-stage, simple and controllable process is implemented.

Very thin layers of nanosilicon are deposited using IL due to the low electrical conductivity of silicon at room temperature [86]. The main advantage of ILs in comparison with aqueous solutions is a wide window of electrochemical stability. As a result, a stronger polarization of the electrode can be used. This makes it possible to conduct electrodeposition of electronegative metals (Al, Mg, Ta, Nb, Mo, W) and non-metals. Katayama et al. obtained a very thin layer of silicon using 1-ethyl-3-methylimidazolium hexafluorosilicate at 90 °C [87]. However, the silicon layers obtained in this way are completely converted into silicon dioxide after contact with air. The possibility of obtaining electrodeposited silicon from SiCl_4_ on the gold surface was demonstrated by voltammetry using 1-butyl-1-methylpyrrolidinium bis(trifluoromethanesulfonyl)imide as an electrolyte, i.e., an ionic liquid that is stable with respect to both air and water [88,89,90]. In this case, a layer with a thickness of several hundred nanometers was formed, which consisted of spherical crystallites with a diameter of 50–200 nm. Silicon nanoparticles with a size of 100–500 nm were also obtained on the copper surface using an ionic liquid and propylene carbonate as a mixed electrolyte [91]. In some cases, the formation of silicon alloys with the substrate metal was detected.

Tsuyuki et al. [92] reported the preparation of silicon nanofilms in *N*,N,N-trimethyl-N-hexylammonium bis(trifluoromethanesulfonyl)imide under light irradiation; SiCl_4_ was used as a precursor. Silicon nanofilms doped with aluminum were also obtained by adding AlCl_3_ to the system.

It was shown [93] that Si can be electrodeposited as a thin film and crystals using the ionic liquids BMImTf_2_N and BMImPF_6_ (with a substantial water content), respectively, on a HOPG substrate [93]. The influence of quaternary ammonium bases was studied by Nicholson [94]. Quaternary ammonium salts (chlorides and bromides) with alkyl groups of different lengths (Et, Pr, Bu) were selected for the study. At the same time, nanolayers with a thickness of about 100 nm were obtained, but these were contaminated with oxygen, chlorine, bromine, hydrogen and phosphorus. Materials with silicon doped with aluminum were also synthesized.

Porous silicon precipitates were obtained in polypropylene carbonate and tetrabutylammonium chloride from SiCl_4_ [95]. Silicon nanofilms were also prepared using Ni, Ag, Pt, Au and glass carbon as substrates, as well as tetrabutylammonium chloride as an electrolyte [96]. Ionic liquids based on the imidazolium cation and amorphous porous silicon nanoparticles are a new hybrid electrolyte with unique controllable electrochemical properties and an exceptionally wide window of redox stability [97]. It should be noted that in most of the described works, the obtained films were contaminated with a significant amount of other elements introduced from the electrolyte.

## 5. Metal Nanoparticles

The number of works on the electrochemical synthesis of metal nanoparticles in ionic liquids is limited compared to the number of studies on oxides. At the same time, the first stage of obtaining oxide nanostructures on the surface of metals under conditions of anodic polarization, including the formation of hexagonal cells, apparently represents the process of redistribution of metal atoms on the surface and self-organization into hexagonal structures. A few examples of the synthesis of metal nanoparticles (1D structures) by electrochemical methods, primarily by the electrodeposition of metal nanowires and nanorods with the control of the morphology of the structures formed (shape-controlled electrodeposition) are presented in the recent review [98]. Synthesis can be carried out both in the absence of a template and in the presence of both soft and hard templates (meaning the flexible or rigid structure of the template molecule). The peculiarities of the method are associated with the use of micellar electrolytes or microemulsions as electrolytes, while it is possible to obtain mesoporous nanostructures with an increased external surface area. The same authors obtained bimetallic Co–Pt nanorods characterized by a developed mesoporous texture [99]. The systems “water in ionic liquid” and “ionic liquid in water” were used as microemulsions. This approach can be applied to obtain both 1D structures containing only one metal and to bimetallic systems; in the latter case, a synergy of the properties of metals is possible, which expands the possibilities of their practical application. Thus, structures of the “core–shell” type (core@shell nanowires/nanorods) were synthesized. Subsequent processing of these materials (surface oxidation, electroplating) allowed the formation of an active shell or layer on the surface of the nanostructure, as well as the variation of the biocompatibility or hydrophilicity/hydrophobicity of the resulting nanomaterials. It is possible to obtain materials for biomedical purposes or biomedical nanorobots, as well as hybrid structures, for example, containing nanowires with nanoparticles of a different nature localized at the ends of the nanowire or bio-organic hybrid nanomaterials. Transition metal particles were synthesized from different precursors in various ionic liquids [100]. In this case, the ionic liquid acts not only as an electrolyte, but also as a stabilizer of metal nanoparticles.

Electrodeposition of aluminum in 1-ethyl-3-methylimidazolium bis(trifluoromethylsulfonyl)imide and 1-ethyl-3-methylimidazolium chloride was described [101]; AlCl_3_ was used as a precursor. In the electrochemical deposition of aluminum from the ionic liquid [C_2_MIm][Cl]-AlCl_3_, partial decomposition of the cation is a necessary condition for obtaining a nanocrystalline metal deposit [101]. Using pyrrolidinium ionic liquids, a nanocrystalline metal deposit is formed, and microcrystalline metal is formed in imidizolium ionic liquids, while the particle size increases with an increase in the deposition temperature. Both effects can be explained by a change in the adsorption of cations on the electrode surface [102,103]. The smallest particles (20 nm) were obtained using trihexyl-tetradecyl-phosphonium bis(trifluoromethylsulfonyl)imide (P_14,6,6,6_ Tf_2_N).

An unusual result obtained by electrochemical deposition of aluminum on uranium from the ionic liquid EMIMCl-AlCl_3_ by galvanic substitution (galvanic displacement) is described [104]. In this case, dendritic aluminum particles are obtained, which are separated from the surface of the uranium electrode by a dense nanoscale layer of aluminum. The electrodeposition of Ag, Cd, Cu and Sb in [BMIM][BF_4_] or [EMIM][BF_4_] has been described by some researchers [105,106,107,108]. Nickel and iron nanoparticles with a size of about 3 nm can be electrochemically synthesized from their triflates using a triflate ionic liquid [109]. Interestingly, the addition of acetonitrile to the electrolyte leads to a decrease in the particle size from 3 nm to 1 nm. The electrodeposition of nickel, cobalt and their alloys with aluminum on the surface of Au(111) was also studied in an earlier work [89].

The electrochemical modification of the surface of sintered metal FeCrAl fibers in ionic liquids BMIM-NTf_2_ and BMIM-Ac has been studied in detail. It is shown [110] that at moderate processing time and anodic currents, a distorted hexagonal cellular structure is formed on the surface (Figure 5). The distortion of the hexagonal structure is apparently caused by the shielding of the electric field due to the fibrous structure of steel. Fibers with such a surface structure were applied as substrates of effective catalysts for the partial oxidation of methane [111].

Sun et al. reported the synthesis of nanoporous structures based on metals (Au, Pt, Ag) by electrodeposition of binary alloys (Au-Zn, Pt-Zn, Ag-Zn) followed by electrochemical removal of one of the alloy components, i.e., zinc (dealloying), using a mixture of zinc chloride and EMIMCl [112,113,114]. The possibility of the repeated use of the ionic liquid is shown, while zinc is included in the recycling (the formation of the alloy at the first stage and its removal from the alloy at the second stage).

There are published works on the electrochemical production of copper nanoparticles in ionic liquids, because the method is relatively simple, cheap, and environmentally friendly, and copper nanoparticles are formed with a high yield and low energy consumption [115,116,117,118].

Titanium nanowires were obtained by electrodeposition from TiCl_4_ in BMIM-NTf_2_ on Au(111), while the formation of titanium nanoparticles with a size of 1–2 nm and Ti-Au alloy particles was noted [119,120].

The deposition of tantalum is a complicated task that cannot be performed using aqueous solutions. Electrochemical deposition of tantalum is possible only in ionic liquids in the form of thin films of nanometer thickness with the use of TaF_5_ as a precursor and 1-butyl-1-methylpyrrolidinium bis(trifluoromethanesulfonyl)imide as an electrolyte. The introduction of LiF into the solution contributes to an increase in the deposition efficiency [121].

Ionic liquids can also be used to produce more complex composite nanomaterials based on metal nanostructures. This goal was achieved by electrolysis of TiCl_4_ in BMIM-NTf_2_ on a special electrode based on highly oriented pyrolytic graphite [119]. According to in situ scanning tunneling microscopy data, titanium was deposited on the terrace steps in the form of nanowires with a diameter of 10 nm and a length of up to 100 nm.

The synthesis of Ag and Ag–Cu nanoscale coatings on glass carbon in triethylammonium acetate by electrodeposition was described [122]. Studies of cyclic voltammograms indicate that the silver deposition current decreases with a shift of the cathode and anode potential peaks to a more positive region with an increase in the copper content. At the same time, the difference in the potentials of the beginning of deposition of Cu and Ag decreases, which favors a more efficient simultaneous deposition of both metals. It should be noted that the morphology of the deposited nanomaterial changes significantly–from a cubic structure to a dendritic one (in the form of flowers) with an increase in the copper content in the material. The Ag-Cu composite containing 23% Cu (Ag77–Cu23) demonstrates the widest window in the cathode potential range when a mixed electrolyte-N,N-dimethylformamide-tetrabutylammonium tetrafluoroborate is used.

Mesoporous bimetallic Co-Pt nanorods with pronounced magnetic properties were first synthesized in nanoscale channels of polycarbonate membranes using microemulsions in ionic liquids [99]. The obtained materials exhibit high electrocatalytic activity in the oxidation of methanol, exceeding 12 times the activity of dense platinum nanorods or supported Pt/C catalysts. This result makes mesoporous bimetallic Co-Pt nanorods promising for use in methanol fuel cells [98].

Tin nanowires are obtained by electrochemical deposition from solutions of SnCl_4_/ionic liquid (BMP-TFSI) containing SiCl_4_ on various substrates, including glass carbon, as well as Cu, Al, and Sn foil [123]. It was found that SiCl_4_ promotes the growth of nanowires having a diameter of 30 nm and a length of 90 microns.

The review by Rudnev [124] describes the use of ionic liquids for the production of lanthanides and their alloys. It is established that Eu and Yb are not electrochemically reduced to metals.

Based on all the above, it can be concluded that for the successful targeted synthesis of metal and alloy nanoparticles in ILs, it is necessary [125]:To investigate in situ the effects of the modulating factors, including cations, anions, metal salts, water content and additives on the nucleation and growth in ILs; such methods as in situ SEM and in situ TEM are efficient.To deeply and systematically study the variation in the interfacial structure of the ILs-electrified substrate with modulating factors, including the nature of the cations and anions, metal salts, water content and additives by a combination of experiments and simulation methods.To precisely characterize the influence of anions, metal salts, water content and additives on the process, with spectroscopic methods such as UV/Vis spectroscopy, Raman spectroscopy and extended X-ray absorption fine structure (EXAFS) being recommended for the study.To find the relationship between the modulating factors and the quality of the deposited materials, enabling the easy tuning of electrodeposition in ILs.

## 6. Nanomaterials Based on Oxides

A comparison of the production volumes of oxide nanoparticles shows that titanium dioxide (TiO_2_) is produced the most, at a volume of 10,000 tons/year. Other common nanoparticles are cerium oxide (CeO_2_), iron oxide (FeO), aluminum oxide (AlOx) and zinc oxide (ZnO), with a production volume of 100 to 1000 tons/year [7].

Usually, for the synthesis of functional semiconductor nanomaterials based on metal oxides, reactions occurring at high temperatures are used, such as laser irradiation, ion implantation, chemical deposition from a vacuum or thermal decomposition. Ionic liquids can be effective in these syntheses while reducing the solvent consumption. In recent years, ionic liquids have been used to produce and stabilize oxide nanoparticles [48,55,56,59]. There are known works on the electrochemical synthesis of zinc oxide nanoparticles from a solution of zinc acetate in choline-chloride-based eutectic mixtures with ethylene glycol and urea (DES) [126]; the particle size is about 12–13 nm, if the synthesis is carried out in the presence of tetrabutylammonium bromide or polyvinylpyrrolidone to stabilize the resulting particles. Without these additives the particle size reaches 18 nm.

Electrochemical preparation of nickel oxide nanolayers from nickel acetate in [BMIM][NTf_2_] with a fairly high specific surface area (about 140 m^2^/g) and a high capacity (about 200 F/g) is described in [127]. On the contrary, NiO nanoparticles with a size from 1 to 3 nm are obtained using [BMP][NTf_2_] and nickel triflate as a precursor.

The synthesis of titanium(IV) oxide nanoparticles of various sizes and shapes (spherical particles, nanotubes, and nanorods) in ionic liquids has been described in numerous studies, and the key role of water additives in this process has been established [128,129,130,131,132]. The largest number of works on the electrochemical synthesis of nanostructures based on metal oxides using ionic liquids are devoted to titanium dioxide nanotubes formed under conditions of anodic polarization [133,134,135]. Titanium oxide nanostructures find diverse applications, for example, in photocatalysis, solar energy harvesting and conversion, sensors, construction materials [7] and for radio-frequency ceramics. Of the three main polymorphic modifications of titanium oxides (anatase, brookite, rutile), until now, research has focused on the synthesis of anatase nanoparticles. Nanometer-sized rutile is attracting more and more attention due to its promising potential for use as a photocatalyst and as an electrode material. It is known that anatase with a particle size of 10–20 nm is a thermodynamically stable modification of TiO_2_. The synthesis of titanium dioxide nanocrystals (2–3 nm) and their self-assembly into mesoporous TiO_2_ nanospheres was carried out in an ionic liquid under mild conditions. The resulting nanostructures combine the advantages of larger spheres with a high specific surface area and a narrow pore size distribution and are expected to have huge potential in solar energy conversion, catalysis and optoelectronic devices [136]. However, the mechanism of formation of TiO_2_ nanotubes remains insufficiently studied. According to one theory, the formation of an oxide film is determined by the ion current, and the electron current causes the release of gaseous oxygen [137]. Based on this theory, a three-stage mechanism for the formation of titanium and zirconium oxide nanotubes was proposed [138]. The formation of nanotubes on the Ti surface in various electrolytes containing 0.7 wt % NH_4_F and 1.8 vol. % H_2_O was studied [139]. It was noted that nanotubes are formed in ethylene glycol, whereas dense TiO_2_ layers are formed in ethanol. These results raise questions about the mechanism based on the action of fluoride ions.

In the typical synthesis of TiO_2_ nanoparticles, ionic liquids 1-butyl-3-methylimidazolium tetrafluoroborate [BMIm][BF_4_] and 1-butyl-3-methylimidazolium hexafluorophosphate [BMIm][PF_6_] were used as solvents [140,141]. Titanium oxide nanotubes with an adjustable pore size were obtained in ionic liquids by varying the electrochemical parameters [142]. Titanium oxide nanotubes with a diameter of 30 to 45 nm were obtained. Water or propylene glycol was added to the solvent (ionic liquid) [143]. The Ti^4+^ ions formed during the anodic dissolution of titanium in the presence of water form aqua complexes of the composition [Ti(H_2_O)_4_]^4+^, which produced titanium oxide during hydrolysis. Nanoscale nickel oxide rods were obtained in a similar way [144]. Glycols and polyols are important additives in the synthesis of well-characterized nanostructured oxide materials, since they are chelating ligands with respect to metal ions, which allows the control of the rate of hydrolysis of transition metal alkoxides [145]. Glycolates of the composition Ti(OCH(CH_3_)CH_2_OH)^2+^ are formed in situ [143]. Figure 6 shows a micrograph of a fragment of the surface of a titanium anode obtained when exposed to an anode current in the ionic liquid BMIMCl in the presence of a water additive in an air atmosphere at a temperature of 25 °C [143]. The fibrous structure of titanium oxide is clearly visible in Figure 6. The length of the fibers can be from 0.1 to several microns; the thickness of the fibers corresponds to approximately 100 nm.

Electrochemical synthesis of titanium dioxide nanopowders on the surface of metallic titanium was carried out in choline-chloride-based eutectic mixtures with ethylene glycol and urea (DES) [133]. The efficiency of synthesis is 92%. The resulting TiO_2_ powder with a nanoparticle size of 8–18 nm is amorphous, and subsequent calcination at 400–600 °C produced anatase.

In a number of works [37,146,147,148,149,150], the process of formation of oxide nanostructures on the surface of metal electrodes (titanium, nickel, stainless steel) was studied in detail. Optimal galvanostatic and potentiostatic modes of formation of nanostructures on the surface of nickel and stainless steel are chosen (Figure 7). The dynamics of the formation of cellular structures of the Benard cell type has been studied. The growth of the “walls” of the cells in height is noticeable in the duration of the anode effect. The cells, perhaps, later serve as the basis for the formation of a layer of nanotubes.

Nanotubes are formed on the surface of titanium in the presence of propylene glycol added to BmimCl (1:1) (Figure 8a). The length of the nanotubes is about 200 nm, the outer and inner diameters are 50 and 20 nm, respectively. EDMA data (Ti, 61–63; O, 24–27; C, 7–8; Cl, 4–5 at. %) showed that the resulting nanotubes are titanium oxide TiO_2_ [150].

In order to obtain a more ordered layer of an oxide film consisting of nanotubes, a nickel plate with an ordered cellular structure previously formed on the surface was placed in a solution of BMImCl-propylene glycol (1:1) and anodically polarized. The best results were obtained at i = 7.5 mA/cm^2^ and t = 100–300 s (Figure 8b). The oxygen content on the surface, according to EDMA data, indicates the formation of a thin layer of nickel oxide: Ni, 55–78; O, 22–45 at. % [144].

It can be assumed that the formation of a film consisting of metal oxide nanotubes occurs through the formation of a cellular structure (nanostructured substrate) on the surface and in the presence of auxiliary substances in the electrolyte that are a source of oxygen.

Water plays a significant role in the anodic electrochemical synthesis of nanostructured oxide materials, since it can change the morphology of titanium oxide at the anode from loose oxide to nanostructured oxide (nanopores, nanorods, nanotubes). The data obtained during the anodic electrochemical treatment of the titanium anode in the medium of both hydrophilic BMImCl and hydrophobic BMImNTf_2_ ILs indicate that titanium oxide nanoparticles can be formed on the surface of titanium in the form of nanospheres or nanofibers, depending on the conditions of synthesis [147].

The metal surface is covered with a “natural surface oxide” as a result of interaction with oxygen in the air. The method of electrochemical transients can provide information about the thickness of the natural surface oxide. Data were obtained on the thickness of the natural surface titanium dioxide, which did not exceed 7 nm. Apparently, this dense oxide prevents the formation and growth of nanostructures to a certain extent. In order to “open” and destroy the dense oxide film, an increased current density value was used for several seconds before the main synthesis. This activation method leads to the formation of nanostructures of a different type, in the form of nanorods (soldered nanotubes) in a relatively short time and at an insignificant current density.

Thus, it can be concluded that the anodic formation of amorphous nanostructures of titanium dioxide and other metal oxides in the studied IL is possible only in the presence of a sufficient amount of water. In the absence of water, only a layer of unstructured anode oxide is obtained [147]. As determined by the Gunterschulze–Betz equation, the growth of the oxide at a constant potential in the absence of F^−^ ions is a self-limiting process, since the electric field gradually decreases with a continuous increase in the thickness of the oxide [151].

The role of water in the formation of oxide nanostructures on titanium under anodic action can be generally represented by the equations [147]:Ti^0^ (metal, surface) − 4e → Ti^4+^ (near-electrode layer)(1)
2H_2_O (additives to IL) + Ti^4+^ → TiO_2_ (nanostructures) + 4H^+^(2)

The addition of propylene glycol to the ionic liquids BMImNTf_2_ and BMImCl in a volume ratio of 1:1 leads to the formation of nanostructures in the form of disordered tubes (Figure 9a) and rods (Figure 9b).

When titanium is oxidized at a constant potential for 10–20 min (Figure 9b), the formation of rods with a diameter of 150–200 nm and a length of 1000 nm is observed in the BMImNTf_2_:propylene glycol mixture. The addition of propylene glycol can increase the solubility of products compared to pure IL, which leads to the formation of various oxide nanostructures on the titanium surface. Nanotubes are mainly formed in the course of galvanostatic oxidation in the presence of propylene glycol (Figure 9a). Thus, the propylene glycol content in the IL and the method of anodic exposure control the morphology of the titanium oxide at the anode from micron-sized rods to nanotubes. Traditionally, it was believed that the key parameters determining the growth and final dimensions of oxide tubes on titanium were the concentration of fluoride ions and the water content in the electrolyte. The mechanism of growth of nanotubes on titanium, even in an aqueous medium, has not yet been sufficiently studied and is debatable. The disadvantages of electrochemical production of oxide nanotubes in an aqueous medium are:the necessity of pretreatment (annealing, polishing, etc.);a two-stage method for producing nanotubes;duration (more than 30 min to obtain primary ordered pores);single use of electrolytes that do not always meet the criteria of “green chemistry.”

These disadvantages, as can be seen from the above results, can be eliminated by the use of ionic liquids as an electrolytic medium. Figure 10 shows photos of the scanning electron microscopy of a fragment of the surface of a titanium foil anodized at the direct current in the BMImNTf_2_ ILs with the addition of propylene glycol [148].

Figure 10c clearly shows areas where only hexagonally ordered structures can be found and at the same time areas with grown nanotubes are observed. The nanotubes have an outer diameter of about 50 nm and a length of about 200 nm. The initial hexagonal structures serve as a matrix for the further growth of nanotubes (Figure 10a,b). The initial nanocells on the surface of titanium are formed in areas with different micro-dimensions, where the current density is higher. Further, ordered structures can be formed under certain conditions due to the instability, which is based on the adsorption of molecules, preferably on the protrusions of the surface (shielding molecules) that prevent dissolution.

An important task is related to establishing a connection between the geometric parameters of the nanostructure on the surface formed during the electrochemical polishing of metal with the conditions of anodic exposure. The formation of hexagonal cells and nanotubes during anodic oxidation in aqueous solutions was initially observed for aluminum, and later it was discovered for a number of other metals (nickel, titanium, hafnium, niobium, tantalum, tungsten, vanadium, zirconium). All these studies, however, were previously carried out in aqueous solutions, and they were aimed at determining the conditions for the formation of nanostructures, while the mechanism of formation of ordered structures that occur on the surface during electropolishing has not been sufficiently studied.

Previously, a mathematical model describing the criterion for the formation of a structured surface was proposed. This model is based on the Debye–Hückel approximation used for dilute electrolytes [35]. The validity of the Debye–Hückel theory in the case of ILs is a subject of discussion. At the moment, there are experimental results confirming the behavior of ILs as dilute solutions of electrolytes, in particular, the most frequently studied IL, BMImNTf_2_. At high anode potentials, IL anions form a dense adsorption layer in the near-electrode space. Water molecules can participate in the formation of the Helmholtz layer and act as shielding molecules. The qualitative correspondence of our results [148] with the results of Yuzhakov’s work [36] is another argument for considering ILs as a dilute electrolyte in this case.

Thin solid films of almost any inorganic material can be rolled into nanostructures called nanorolls [153]. If the rolls have a large diameter and thin walls, they can be used as nanocontainers or nanocapsules. It was already noted earlier that the surface of metals is covered with a “natural surface oxide”, which plays an important role in the formation of nanostructures [154]. The formation of oxide nanorolls decorated with oxide nanotubes was found [155] (Figure 11a) during the anodic oxidation of an amorphous Fe70Cr15B15 alloy in the ionic liquid BMImBF_4_. This unusual architecture was first observed on the surface of an amorphous alloy. The generation of a new type of nanostructure by electrochemical oxidation of the amorphous Fe70Cr15B15 alloy occurs only in the presence of a natural oxide film on the surface. The conditions and scheme for the formation of such a combined nanostructure are shown in Figure 11b.

## 7. Electropolymerization in Ionic Liquids

### 7.1. Nano-Sized Polymer Films

The synthesis of nanofilms of conductive polymers on conductive substrates using ionic liquids as electrolytes is of interest. The use of ILs as a medium leads to a significant change in the morphology of the films and improvement in their conductive properties. The available reviews [24,156,157,158] describe the synthesis, preparation and use of π-conjugated conducting polymers in IL based on the [BMIm]^+^ cation and BF_4_^−^, PF_6_^−^ and AlCl_4_^−^ anions. Polymers in ionic liquids demonstrate high stability during electrochemical potential cycling.

The authors of this review partially presented the state of the art in this area in the previous work [159]. To date, a large number of studies have been published showing significant progress in this area. Thus, the electrochemical reactions of pyridine in 1-butyl-3-methylimidazolium tetrafluoroborate ionic liquid were investigated using cyclic voltammetry [160]. It was proved that pyridine can be reduced on rhenium and platinum electrodes to an anion radical in ionic liquid. A comparative study of the Re and Pt electrodes was conducted for the first time. The product of the cathodic reduction of pyridine under the action of the anodic potential is further converted into an oligomeric product that is soluble in ionic liquid (Figure 12). A scheme for the formation of a conductive oligomer is proposed (Figure 1a,b). SEM images of rhenium and platinum surfaces after 100 cycles in IL with dissolved pyridine are presented in Figure 13. Conjugated polymers can be doped with cations or anions of the electrolyte. The formation of an oligomeric product doped with cations of ionic liquid is proposed (Figure 1c).

The electrochemical oxidation of benzene, biphenyl and *p*-terphenyl in ionic liquid BPC-AlCl_3_ (1:1.5) at 30 °C was studied by cyclic voltammetry. A comparative analysis of the peaks obtained for each initial monomer was carried out. Assumptions were made about the difference in the mechanisms of electrochemical synthesis of polyphenylenes produced from benzene, biphenyl and *p*-terphenyl in the ionic liquid [161]. Linear polymers with a polymerization degree of 8–120 were synthesized by the electrooxidation of benzene on a platinum and glass carbon electrode. It was shown that the degree of polymerization increases with an increase in the anode potential [162].

A fairly complete bibliography of works before 2010 on electropolymerization in ILs is presented in a review by Heinze et al. [163].

Another review [28] examines the current results related to the electropolymerization of conventional conductive polymers, including choline-chloride-based eutectics. The mechanisms of electrochemical polymerization in the synthesis of polypyrrole, polyaniline and poly(3,4-ethylenedioxythiophene) using various compositions of DES, as well as their characteristics, mainly from the point of view of surface morphology, are considered. It is important to simplify the existing ionic liquid media for the electropolymerization of aniline and further improve the electrochemical characteristics of polyaniline obtained in ionic liquid media. Electrochemical polymerization of aniline was carried out in an ionic liquid (pyrrolidinium hydrosulfate) with high proton activity. It is shown that electropolymerization of aniline proceeds easily in this medium, and the resulting polyaniline has a porous structure and a high specific capacity [164].

Electrochemical polymerization of pyrrole in the ionic liquid BMimPF_6_ was carried out on a smooth glass carbon electrode. The films of polypyrrole showed good adhesion to the electrode surface; the properties of the films were characterized by UV spectroscopy and SEM [165]. A bipolar electrochemical system has recently been used to modify conducting micro- and nanoobjects with different surface layers. Pyrrole was first electropolymerized on a bipolar electrode in ionic liquid [BMIm][TFSI] and in acetonitrile (Figure 2) [166].

The development and production of polymer-based electrochromic devices is considered in the literature [167]. The possibility of using electropolymerization, which leads to the production of conductive polymers for the synthesis of such devices, is noted. Poly(3,4-ethylenedioxythiophene) (PEDOT) was obtained for the first time in the process of one-stage electropolymerization using the ionic liquid 1-ethyl-3-methylimidazolium hydrosultate (EMIMHSO_4_), and the use of the polymerization product as a counter-electrode in dye-sensitive solar cells (DSSC) was shown [168].

The use of conducting polymers in combination with ionic liquids is widely considered [169]. The review focuses on the following specific applications: energy (energy storage devices and solar panels), use in environmental purification (capture of anions and cations, electrocatalytic reduction/oxidation of pollutants on electrodes based on conductive polymers, adsorption of pollutants) and, finally, electroanalysis as chemical sensors in solution, the gas phase and the detection of chiral molecules. The original method of producing polyethylene in an ionic liquid on nanostructured titanium dioxide by electrochemical reduction of CO_2_ was recently demonstrated [170]. Unfortunately, these studies have not been continued.

### 7.2. Electropolymerization of Ionic Liquids

Polymerized ionic liquids or poly(ionic liquids) (PILs) belong to a special type of polyelectrolytes that preserve the properties of ILs in each repeating structural unit. The main advantages of using PILs instead of ILs are their increased mechanical stability, durability and spatial controllability compared to ILs [171]. The possibility of using membranes obtained by incorporating the IL EMIM NTf_2_ into a polymer matrix (polymer ionic liquid, PILs) for separating a mixture of CO_2_ and CH_4_ is demonstrated in [172]. Obtaining PILs by electropolymerization has a number of advantages compared to chemical polymerization, including the easy generation of thin polymer films on various surfaces and the ability to modify and/or create surfaces using redox active molecules. A series of imidazolium-based ionic liquids containing a terminal pyrrole fragment was synthesized and electrochemically polymerized. Polypyrrole films with an unprecedented folded structure, even hierarchical structures, can be obtained by the polymerization of functionalized pyrrole monomers without using any template. In addition, their surface properties can be easily corrected without losing the electrical and optical properties and morphology of the polypyrrole films with the help of a simple anion exchange reaction [173]. It was found that the polymerization ability of synthesized ionic liquids strongly depends on the type of anions. Although the bromide monomer does not polymerize, well-characterized polymer films can be formed on various substrates in the case of anions containing BF_4_^−^ and PF_6_^−^.

The potentiodynamic electro-oxidative polymerization of a bifunctional monomer with both vinyl and thienyl groups attached to ionic liquid based on alkylimidazolium is demonstrated (Figure 3).

The resulting polymer film adheres to the surface of the electrode and, when examined in polarized light, turned out to be optically birefringent [174].

Coatings of a conductive polymer ionic liquid (PIL) were obtained on various electrodes by the electropolymerization of [pyrrole-C6MIm]PF_6_. Electrochemical studies have shown that the polymer is more selective to anions and acts as an electrostatic repulsion barrier for cations [175]. Poly[thiophene-C_6_MIm][NTf_2_] films exhibit high thermal stability and can be used as sorbents for solid-phase extraction [176].

The analysis of the polymerization product is usually carried out using such methods as NMR spectroscopy, high–resolution mass spectrometry (HRMS), infrared Fourier spectroscopy (FTIR) and thermogravimetric analysis (TGA). The morphology and thickness measurements of the deposited films are evaluated using a scanning electron microscope. These methods allow one to characterize the properties of films, but the degree of polymerization is not determined in these works.

## 8. Nanoscale Composites

Attempts to obtain various composite materials by electrochemical methods using ionic liquids are interesting. The electrochemical synthesis of electrically conductive polymer composites offers a number of advantages in comparison with chemical synthesis: the ability to include various alloying additives, simple control of microstructures, sequential deposition to obtain layered structures and the ability to form copolymers [177]. Nanocomposites based on a polymer ionic liquid and metal nanoparticles (poly(MImEO8BS)-Ni) were obtained for the first time on glass carbon [178,179]. The obtained materials were used in the processes of the electrochemical oxidation of some biologically active substances (drugs).

## 9. Conclusions

The field of research on ILs has reached a significant level due to the variety of combinations of ions with different abilities for self-assembly, both in the volume of the liquid and in the boundary medium [51]. Their chemical and self-organized structures can be used in many areas of scientific research and practical applications. Thus, ionic liquids can be successfully used to produce various nanoscale structures on the surface of conductive materials, including semiconductor materials, oxide nanotubes, nanorods, metallic and bimetallic nanoparticles, including “core-shell” structures, as well as films and two-dimensional coatings of nanoscale thickness and nanocomposite and hybrid nanomaterials. It should be recognized that in some cases these processes are complicated by the contamination of the obtained nanomaterials with ions and elements that are part of the ionic liquid. In addition, when ionic liquids are used as solvents, gas absorbents or electrolytes, reactions between ILs and substrates can lead to the formation of by-products [180]. For this reason, reactions should be carried out with extreme caution, especially in the presence of catalysts with a low content of transition metals. The by-products obtained from the ionic liquid as a result of these reactions change the chemical and physical properties of the ionic liquid, such as its viscosity, conductivity, solubility and adsorption capacity, and these changes complicate the regeneration of ionic liquids [181].

A review [182] noted that among the reasons limiting the use of nanomaterials, one can single out the relatively high cost of their synthesis. The cost of ionic liquids is one of the reasons limiting the possibility of their use. The formation of nanostructures on the electrode surface may be an undesirable process. For example, the uneven deposition of metal and the formation of dendrites in high-density energy storage devices reduces the efficiency, safety and service life of batteries with metal anodes. Ultra-concentrated ionic liquid electrolytes (for example, ionic liquid 1:1: alkali metal ion) in combination with pre-conditioning at more negative potentials can completely eliminate these problems and, consequently, revolutionize energy storage devices with high density [183]. A wide window of electrochemical stability of many ionic liquids allows one to obtain nanomaterials that cannot be synthesized from aqueous solutions or in non-aqueous electrolytes and solvents.

In the liquid state, ionic liquids can form extended systems due to hydrogen bonds. In this sense, ionic liquids can be considered “supramolecular” solvents. This special property can be used in the synthesis of extended ordered nanoscale structures or for controlling morphology using the template effect of various ionic liquids [67,184].

As was noted, ILs have already established themselves as [41]:designer solvents for various processes;media for electrochemical reactions, catalysis [185], electrocatalysis and electrodeposition;electrolytes for power sources and generators, such as supercapacitors, and batteries for solar cells or fuel cells (ILs with high proton conductivity have been found);liquids for electric wetting;liquid organic components for self-assembled nanoplasmon devices;electronic valve media for monomolecular devices and electrolytes for electronics and superconductors;media and electrolytes for chemical and electrochemical sensors.

## Data Availability

Not applicable.

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
