# Peer review of "Electrochemical Synthesis of Unique Nanomaterials in Ionic Liquids"

_nanomaterials, 2021, doi:10.3390/nano11123270_

Round 1

Reviewer 1 Report

The authors wrote: This model is based on the Debye-Hückel approximation used for diluted electrolytes. It would be helpful for the authors to explain the application of DH theory to the electrodeposition process. I believe that most readers are not familiar with that.

At E1 > 0, the solubility increases on the protrusion… What is E1?

In a sentence: When using suitable ionic liquids (IL), reagents and products that are unstable in 210 other solvents can become stable and redox processes that do not occur in an aqueous medium are possible… I suggest that aqueous media replace with molecular solvents or add the words organic solvents after aqueous media.

Page 6, line 234: Did authors mean: 1,3 dilalkylimidazolium?

For phosphonium and 235 quaternary ammonium ions, the cathode limit is -2.7 V. The oxidation stability of anions 236 can be represented as: [TFSA]- > [FAP]- > [TfO]- > [DCA]- > [TFA]-. Does it depend on the type of electrode?

Page 6, line 245: Should be changed to: degree of structural organization of ionic liquids plays…

Page 6: The authors could add 1-2 sentences about the possibility and limitation of DES in electrochemical synthesis.

Page 6, line 268: I think the word bis-triflateimide is misspelled.

Depending on the conformation of the butyl group in the [C4MIm]+ cation, IL can be 265 crystallized either in a monoclinic (trans-trans) or orthorhombic forms. The authors should state the source of this information, it is not clear what trans-trans refers to?

Page 7, lines 290-314, the presence of any image would be valuable to readers.

Page 8, line 351: TBA cation is not mentioned in Table 1.

The authors refer to the NTf2 anion as bis-trifluorosulfonylimides or (triflate) imides or Bis(trifluoromethanesulfonyl)imide, which should be standardized in the manuscript. Bis(trifluoromethylsulfonyl)imide is correct.

page 15, line 666: Authors should specify the composition of the mixture? Choline chloride is not an ionic liquid.

Page 16, line 772: The stoichiometry in reaction 2 is incorrect.

Author Response

Response to Reviewer 1 Comments

The authors wrote: This model is based on the Debye-Hückel approximation used for diluted electrolytes. It would be helpful for the authors to explain the application of DH theory to the electrodeposition process. I believe that most readers are not familiar with that.

At E1 > 0, the solubility increases on the protrusion… What is E1?

Inaccuracy was admitted, this was corrected. It should be explained that the proposed model refers to anodic processes (not electrodeposition, as the respected reviewer believes). We give an excerpt from our article [37] in a separate file (please see "author-coverletter-15597681.v1.pdf"). We believe that it may not be necessary to supplement this review with the mathematical constructions, which the interested reader can always find in the full version of this article ref 37. If a respected reviewer sees the usefulness and appropriateness of this new section (or part of it) in our relatively highly specialized review, then we will be happy to expand it.

In a sentence: When using suitable ionic liquids (IL), reagents and products that are unstable in 210 other solvents can become stable and redox processes that do not occur in an aqueous medium are possible… I suggest that aqueous media replace with molecular solvents or add the words organic solvents after aqueous media.

OK, this was corrected.

Page 6, line 234: Did authors mean: 1,3 dilalkylimidazolium?

OK, this was corrected.

For phosphonium and 235 quaternary ammonium ions, the cathode limit is -2.7 V. The oxidation stability of anions 236 can be represented as: [TFSA]- > [FAP]- > [TfO]- > [DCA]- > [TFA]-. Does it depend on the type of electrode?

These data were obtained for glassy carbon working electrode (ref 44). This clarification was added in the text.

Page 6, line 245: Should be changed to: degree of structural organization of ionic liquids plays…

OK, this was added.

Page 6: The authors could add 1-2 sentences about the possibility and limitation of DES in electrochemical synthesis.

OK, this was corrected. A special Subsection was added at the end of Section 2.

Page 6, line 268: I think the word bis-triflateimide is misspelled.

OK, this was corrected. All spelling variants are replaced by the one mentioned in Table 1 - bis(trifluoromethanesulfonyl)imide.

Depending on the conformation of the butyl group in the [C4MIm]+ cation, IL can be 265 crystallized either in a monoclinic (trans-trans) or orthorhombic forms. The authors should state the source of this information, it is not clear what trans-trans refers to?

Ref 50: “Monoclinic (trans−trans) or orthorhombic (gauche− trans) crystals of these C4mim+ ILs are possible, depending on butyl group conformations. Complementary Raman spectroscopy data revealed an equilibrium of both structures was present, and that interconversion between them may hinder crystallization and thus lower the mp below 100 °C.”. Ref 50 was added in the text and the superfluous clarification (trans - trans) was deleted.

Page 7, lines 290-314, the presence of any image would be valuable to readers.

OK, this was corrected. We agree and have added two new figures (Figures 2 and 3).

Page 8, line 351: TBA cation is not mentioned in Table 1.

OK, this was corrected. The table has been updated.

The authors refer to the NTf2 anion as bis-trifluorosulfonylimides or (triflate) imides or Bis(trifluoromethanesulfonyl)imide, which should be standardized in the manuscript. Bis(trifluoromethylsulfonyl)imide is correct.

OK, this was corrected. All spelling variants are replaced by the one mentioned in Table 1 - bis(trifluoromethanesulfonyl)imide.

page 15, line 666: Authors should specify the composition of the mixture? Choline chloride is not an ionic liquid.

OK, this was corrected. The authors call DES based on choline-chloride as “ionic liquids”. Mixtures choline-chloride with urea and ethylene glycol were used in the work [128].

Page 16, line 772: The stoichiometry in reaction 2 is incorrect.

OK, this was corrected.

Reviewer 2 Report

This review is a well-documented overview of field literature. I have no criticisms upon the choice of references. The manuscript is well-written and pleasant to read. I recommend its publication after consideration of following points.

  1. On lines 247-251, authors evacuate the question of deep eutectic solvent (DES) just saying that DES and IL have many common characteristics and that both terms are often used interchangeably. Fortunately, they also add that some authors consider that IL and DES are two different types of solvents.

This is too short. IL is an organic molecule possessing an ionic part while DES is a salt mixture. Both solvent types have different features, for instance solubility. Usages are different.

A part of the reference reviewed in this manuscript concern in fact DES. So, it is important that the manuscript involves at this place a paragraph clarifying the differences between IL and DES and commenting their common and different features.

  1. When the text refers to DES and not to IL, it should be expressively mentioned, for instance at lines 608, 608, 665, 849…
  2. There are 6 occurrences of "bis(triflate)imide", which is not the usual short name of this anion. Authors can use it, providing that they indicate full name "bis(trifluoromethane)sulfonimide" at first occurrence.
  3. Line 344: "Debay length" => "Debye length"

Author Response

Response to Reviewer 2 Comments

This review is a well-documented overview of field literature. I have no criticisms upon the choice of references. The manuscript is well-written and pleasant to read. I recommend its publication after consideration of following points.

We thank the reviewer for his careful reading of our manuscript, valuable comments and high praise. Of course, all the recommendations we have taken into account, and the comments have been corrected.

  1. On lines 247-251, authors evacuate the question of deep eutectic solvent (DES) just saying that DES and IL have many common characteristics and that both terms are often used interchangeably. Fortunately, they also add that some authors consider that IL and DES are two different types of solvents.

This is too short. IL is an organic molecule possessing an ionic part while DES is a salt mixture. Both solvent types have different features, for instance solubility. Usages are different.

A part of the reference reviewed in this manuscript concern in fact DES. So, it is important that the manuscript involves at this place a paragraph clarifying the differences between IL and DES and commenting their common and different features.

OK, this was corrected. A special Subsection was added at the end of Section 2.

  1. When the text refers to DES and not to IL, it should be expressively mentioned, for instance at lines 608, 608, 665, 849…

OK, this was corrected.

  1. There are 6 occurrences of "bis(triflate)imide", which is not the usual short name of this anion. Authors can use it, providing that they indicate full name "bis(trifluoromethane)sulfonimide" at first occurrence.

OK, this was corrected. All spelling variants are replaced by the one mentioned in Table 1 - bis(trifluoromethanesulfonyl)imide. This is a synonym for the name proposed by the reviewer. Both names are found in catalogs :"bis(trifluoromethane)sulfonamide" and "bis(trifluoromethanesulfonyl)imide". In the electrochemical publications, the second name is more often used.

  1. Line 344: "Debay length" => "Debye length"

OK, this was corrected.

Round 2

Reviewer 1 Report

The authors responded correctly to all my suggestions, so I suggest publishing the manuscript in a presented form.